# DNA Repair and Mutagenesis of ADP-Ribosylated DNA by Pierisin

**DOI:** 10.3390/toxins16080331

**Published:** 2024-07-26

**Authors:** Masanobu Kawanishi, Takashi Yagi, Yukari Totsuka, Keiji Wakabayashi

**Affiliations:** 1Environmental Molecular Toxicology, Department of Biological Chemistry, Graduate School of Science, Osaka Metropolitan University, 1-2 Gakuen-cho, Naka-ku, Sakai 599-8570, Japan; t.yagi@omu.ac.jp; 2Department of Environmental Health Sciences, Hoshi University, 2-4-41 Ebara, Shinagawa-ku, Tokyo 142-8501, Japan; totsuka.yukari@hoshi.ac.jp; 3Graduate Division of Nutritional and Environmental Sciences, University of Shizuoka, 52-1 Yada, Suruga-ku, Shizuoka 422-8526, Japan; kwakabayashi@u-shizuoka-ken.ac.jp

**Keywords:** pierisin, nucleotide excision repair, translesion DNA synthesis, mutagenesis

## Abstract

Pierisin is a DNA-targeting ADP-ribosyltransferase found in cabbage white butterfly (*Pieris rapae*). Pierisin transfers an ADP-ribosyl moiety to the 2-amino group of the guanine residue in DNA, yielding *N*^2^-(ADP-ribos-1-yl)-2′-deoxyguanosine (*N*^2^-ADPR-dG). Generally, such chemically modified DNA is recognized as DNA damage and elicits cellular responses, including DNA repair pathways. In *Escherichia coli* and human cells, it has been experimentally demonstrated that *N*^2^-ADPR-dG is a substrate of the nucleotide excision repair system. Although DNA repair machineries can remove most lesions, some unrepaired damages frequently lead to mutagenesis through DNA replication. Replication past the damaged DNA template is called translesion DNA synthesis (TLS). In vitro primer extension experiments have shown that eukaryotic DNA polymerase κ is involved in TLS across *N*^2^-ADPR-dG. In many cases, TLS is error-prone and thus a mutagenic process. Indeed, the induction of G:C to T:A and G:C to C:G mutations by *N*^2^-ADPR-dG in the hypoxanthine phosphoribosyltransferase gene mutation assay with Chinese hamster cells and *supF* shuttle vector plasmids assay using human fibroblasts has been reported. This review provides a detailed overview of DNA repair, TLS and mutagenesis of *N*^2^-ADPR-dG induced by cabbage butterfly pierisin-1.

## 1. Introduction

It has long been believed that ADP-ribosylation primarily modifies proteins. However, recent studies have uncovered its common occurrence in DNA as well [1]. The first DNA-targeting ADP-ribosyltransferases (ART) was found in cabbage white butterfly (*Pieris rapae*) as a cytotoxic toxin, and was thus named pierisin-1 [2,3]. The mechanism of pierisin toxicity involves the catalytic transfer of ADP-ribose from NAD^+^ to the 2-amino group of the guanine residue in target DNA in cells, resulting in the formation of *N*^2^-(ADP-ribos-1-yl)-2′-deoxyguanosine (*N*^2^-ADPR-dG) [4,5]. This process disrupts cellular function, leading to apoptotic cell death. Pierisin-1 is most abundantly expressed in the transition phase from the final larval stage to the pupal stage of the cabbage butterfly. Therefore, pierisn-1 may contribute to the metamorphosis process. This toxin may also play a defensive role against parasitization and microbial infections in the cabbage butterfly [6].

Pierisin-1 produces mono-ADP-ribosylated dG. Such chemically modified DNA, recognized as DNA damage, triggers cellular responses, including DNA repair pathways, to rectify the damage. Several major DNA repair pathways are known, including nucleotide excision repair (NER), base excision repair, mismatch repair, homologous recombination and non-homologous end joining, and these pathways physically remove the damage in a substrate-dependent manner [7].

Besides the DNA repair system, organisms possess the DNA damage tolerance pathways for certain types of damaged DNA substrates. The translesion DNA synthesis (TLS) pathway is one such DNA damage tolerance process that enables the DNA replication machinery to replicate past the damaged DNA template. Specialized translesion polymerases (pols) bypass the damage to facilitate the continuation of DNA replication, but this can potentially introduce incorrect bases, leading to mutations in subsequent rounds of replication. Therefore, the insufficient DNA repair of damaged DNA substrates could result in mutagenesis via the TLS pathway [7].

Importantly, these processes do not occur uniformly across genomic DNA regions or sites; for example, the local sequence context can affect the efficiency and manner of these processes [8]. Mutational signatures are recurrent patterns of base changes that reflect this specificity. The signatures naturally emerge because each particular mutagenic process or DNA modification substance is more likely to affect certain sites in specific contexts more frequently than others [9].

The general property and biological function of pierisin are described in the article by Takahashi-Nakaguchi et al. in the same Special Issue [6]. Instead, in this review, we provide a detailed overview of DNA repair, TLS and mutagenesis of *N*^2^-ADPR-dG induced by cabbage white butterfly pierisin-1. In addition, we discuss the current gaps and potential topics for future studies on the damage response to ADP-ribosylated nucleic acids.

## 2. Nucleotide Excision Repair of ADP-Ribosylated DNA by Pierisin-1

Among the multiple DNA repair pathways that cells possess, NER is a major machinery to remove bulky DNA adducts induced by UV light and chemicals. Within the broad spectrum of DNA lesions repaired by this pathway, significant distortion of the DNA double helix is considered a common feature. NER functions as a simple “cut and patch” system (Figure 1): a short single-stranded DNA fragment containing a lesion is excised by the repair machinery, and then the gap is filled by the DNA synthesis facility to restore the duplex DNA structure [10]. In *Escherichia coli* (*E. coli*), three proteins, UvrA, UvrB and UvrC, collaborate to recognize and excise DNA damage in NER [11]. In humans, defects in NER can cause xeroderma pigmentosum (XP), a hereditary disease characterized by sensitivity to sunlight and susceptibility to cancer. In mammalian cells, the XPA protein binds preferentially to distorted DNA molecules and plays a crucial role as a bridging factor between the initial recognition complex and the excision machinery [10].

In the NER model of *E. coli* (Figure 1), the UvrA dimer loads UvrB onto the lesion site of DNA. This UvrA_2_B complex interacts with the damage and unwinds the local DNA duplex at the lesion. This conformational change in the DNA allows for UvrB to interact directly with the damage, forming a stable UvrB–DNA intermediate. Finally, UvrC interacts with UvrB in the UvrB–DNA complex, and this UvrBC complex incises the DNA strand with the lesion [12].

To understand the structural basis of recognition and incision of *N*^2^-ADPR-dG by these *E. coli* UvrABC proteins, Kawanishi et al. conducted mobility shift gel electrophoresis assays using a 50 mer oligodeoxynucleotide containing a single *N*^2^-ADPR-dG [13]. The results demonstrated that UvrA_2_B proteins preferentially bound to the site-specific *N*^2^-ADPR-dG-modified 50 mer DNA in vitro. Furthermore, the incubation of UvrAB with the modified 50 mer nucleotide resulted in the accumulation of the UvrB–DNA complex, while the incubation of all three UvrABC components yielded the incised product of the *N*^2^-ADPR-dG-modified oligodeoxynucleotide. These data suggest that in *E. coli,* NER is involved in the repair of *N*^2^-ADPR-dG.

In mammalian cells, XPA protein binds to a distorted DNA duplex and functions as a bridging factor between a recognition intermediate and the ultimate endonuclease complex [10]. Therefore, Kawanishi et al. first tested the DNA binding of XPA to an ADP-ribosylated DNA fragment [13]. The results of the filter binding assay and gel mobility shift assay indicated that XPA protein preferentially interacts with pierisin-1-treated DNA fragments in vitro. In the filter binding assay, ^32^P-labeled DNA treated with pierisin-1 was mixed with XPA protein, and then the reaction mixture was run through a filter. If the XPA protein binds to the labeled DNA, radioactivity is retained by the filter. When the XPA protein was incubated with pierisin-1-treated DNA, the radioactivity remained on the filter depending on the amount of XPA protein. They also examined the binding of the GST-fused XPA protein to pierisin-1-treated DNA using the gel electrophoretic mobility shift assay. When GST-XPA was added to the ADP-ribosylated DNA substrate labeled with radioactive ^32^P, upper-shifted bands appeared. As the amount of added GST-XPA increased, the intensities of the shifted bands increased and those of free DNA bands decreased. The intensity of the shifted band decreased when competitor DNA, i.e., non-labeled ADP-ribosylated DNA, was added to the binding mixture. When an anti-GST antibody was added to the mixture, super-shifted bands were observed. These results demonstrate that the XPA protein preferentially binds to pierisin-1-treated DNA, indicating the involvement of NER in the repair of *N*^2^-ADPR-dG in mammalian cells as well.

The involvement of NER in the repair of *N*^2^-ADPR-dG has also been confirmed through a cell viability assay using the NER mutants of Chinese hamster ovary (CHO) cell lines [14]. In eukaryotes, NER consists of two pathways that are distinct in the initial process of damage recognition: repair of lesions across the entire genome, referred to as global genome repair (GGR); and repair of transcription-blocking lesions present in transcribed DNA strands, known as transcription-coupled repair (TCR) [15]. In the excision process of both GGR and TCR, the general transcription factor complex TFIIH, containing XPB and XPD helicases, unwinds a nucleotide duplex at the lesion site. As previously described, XPA verifies the damage in an open DNA conformation and plays a key role in assembling the remaining repair components. Replication protein A stabilizes the opened DNA complex, and XPG and ERCC1-XPF endonucleases incise the nucleotide strand around the lesion. After removal of the excised strand with the lesion, general replication factors fill in the gaps, and DNA ligase forms phosphodiester bonds to seal the nicked DNA. Deficiencies in NER lead to a high sensitivity to DNA damaging agents due to the impaired repair of DNA lesions induced by these agents. Therefore, Shiotani et al. evaluated cell viability after pierisin-1 treatment using NER-deficient cells and their wild-type counterpart cell line [14]. CHO UV5, CHO UV20, CHO UV41 and CHO UV135 cell lines belong to rodent complementation groups 2, 1, 4 and 5, respectively, and are mutated in the hamster homologs of human XPD, excision repair cross-complementation group 1 (ERCC1), XPF and XPG, respectively. They showed that the IC_50_ values of pierisin-1 were 650 ng/mL for AA8 (wild) and 190–260 ng/mL for NER-deficient cells (Table 1). Thus, wild-type AA8 proved to be the most resistant to pierisin-1-induced cytotoxicity. These results also indicate the involvement of the NER system in the repair of *N*^2^-ADPR-dG in mammalian cells.

**Figure 1 toxins-16-00331-f001:**
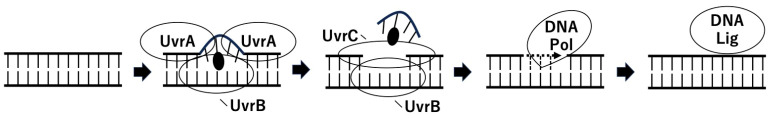
Cut-and-patch model of *E. coli* NER. A short, single-stranded DNA fragment with a lesion (solid ellipse) is excised, and the gap is filled. Here, the UvrA_2_B complex unwinds the DNA duplex at the lesion, UvrC interacts with UvrB, and then UvrC incises the DNA strand with the lesion. Finally, DNA polymerase (DNA Pol) and DNA ligase (DNA Lig) complete the process.

**Table 1 toxins-16-00331-t001:** Sensitivity of NER-deficient CHO cell lines against pierisin-1 [14].

CHO Cell Line	AA8 (Wild)	UV5 (*XPD*^−^)	UV20 (*ERCC*1^−^)	UV41 (*XPF*^−^)	UV135 (*XPG*^−^)
IC_50_ [ng/mL]	650	230	190	260	240

## 3. Translesion DNA Synthesis across Mono-ADP-Ribosylated Deoxyguanosine by Y-Family DNA Polymerases

DNA repair machineries efficiently remove the majority of DNA damage; yet, some lesions may persist and impede DNA replication. Unrepaired damage often leads to mutagenesis and carcinogenesis. Conventional replicative DNA polymerases such as mammalian Pol δ and Pol ε and bacterial Pol I and Pol III possess high fidelity due to their concurrent proofreading 3′ to 5′ exonuclease activity, rendering them incapable of replicating damaged DNA [16]. To address this challenge, organisms have evolved several translesion synthesis polymerases, specialized enzymes capable of accommodating DNA lesions in their active sites and performing TLS (Figure 2). However, this aberrant DNA synthesis is characterized by poor replication accuracy, favoring the formation of non-Watson–Crick base pairs and efficient mismatch extension [17]. In this TLS process, one or more members of Y-family DNA polymerases suited for a lesion bypass typically participate [16]. These polymerases are found across prokaryotes, archaea and eukaryotes, spanning unicellular organisms to humans. The Y-family includes eukaryotic Pol η, Pol ι, Pol κ and Rev1, as well as prokaryotic Pol IV and Pol V [18]. Additionally, B-family Pol ζ can replicate damaged DNA templates in conjunction with some of the Y-family polymerases [19,20,21,22]. In humans, defects in Pol η lead to XP variant [23,24,25]. Crystallographic approaches and biochemical studies have elucidated how Y-family polymerases manage damaged DNA templates [17,26,27,28,29,30,31].

The TLS across the *N*^2^-ADPR-dG conducted by human Y-family DNA polymerases in vitro was reported in 2005 [32]. The study utilized a site-specific modified oligodeoxynucleotide containing a single *N*^2^-ADPR-dG as a template for ^32^P-labeled-primer extension reactions with human DNA polymerases η, ι and κ. The study comprised three experiments: a running start experiment, an insertion experiment and an extension experiment. In the running start experiment, a *N*^2^-ADPR-dG-modified 30 mer template was annealed with “-3 primer” terminating three bases before the *N*^2^-ADPR-dG site. Using this substrate, an elongation reaction was carried out in the presence of all four dNTPs. In the insertion experiment, the “-1 primer” terminating one base before the *N*^2^-ADPR-dG site was annealed to the template, and polymerization reactions were performed with the template–primer substrate in the presence of a single dNTP. In the extension experiment, the ability of the polymerases to extend a mismatched or matched base placed opposite *N*^2^-ADPR-dG was examined. For this purpose, each primer, with A, G, C or T at the 3′-terminus, correctly or incorrectly paired with *N*^2^-ADPR-dG on the templates, was incubated with the polymerases. The results of gel electrophoresis of the polymerization products showed that Pol κ catalyzed efficient TLS past *N*^2^-ADPR-dG by preferentially inserting dC opposite the lesion. Pol κ also extended the DNA strand from a mismatch terminus, where dG, dA or dT had been incorporated opposite the lesion. Pol ι incorporated dCMP and dTMP opposite the lesion, but did not extend beyond it. Pol η preferentially inserted dCMP and three others to a lesser extent opposite *N*^2^-ADPR-dG, and it extended further, albeit with a low efficiency. These results indicate that Pol κ plays a crucial role in bypassing *N*^2^-ADPR-dG.

**Figure 2 toxins-16-00331-f002:**
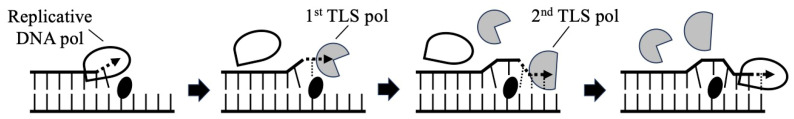
Two-step model of TLS. A replicative polymerase (splash shape) stalls at the site of a DNA lesion (solid ellipse). One TLS polymerase (hatched omnomnomagon) incorporates deoxynucleotides opposite the DNA lesions. This insertion step can be error-prone. And then, the other (hatched semicircle) extends the DNA strand from such deoxynucleotides. Lastly, TLS polymerase is replaced by the replicative DNA polymerase.

## 4. Mutagenesis by Mono-ADP-Ribosylated Deoxyguanosine in Mammalian Cell Lines

Benzo[*a*]pyrene (BP) and tamoxifen (TAM) have been reported to react with the exocyclic *N*^2^-amino group of 2′-dG residues, similar to pierisin-1, resulting in the formation of dG-*N*^2^-BPDE (BP diol epoxide) and dG-*N*^2^-TAM, respectively [33,34,35,36]. These compounds are known to predominantly generate a G to T transversion mutation in mammalian cells or in experiments with oligonucleotides containing a modified single dG [37,38,39,40]. These studies imply that *N*^2^-ADPR-dG also induces mutations.

The hypoxanthine phosphoribosyltransferase gene (*HPRT*) mutation assay with Chinese hamster cells is commonly used in in vitro mammalian mutagenesis studies. Totsuka et al. found that mutation frequencies of *HPRT* gene in pierisin-1-treated Chinese hamster lung (CHL) cells were elevated approximately 40-fold in the assay [41]. Subsequently, a s*upF* shuttle vector plasmid assay [42] was conducted to confirm that *N*^2^-ADPR-dG itself leads to mutations [41]. The shuttle vectors were treated with pierisin-1 and β-NAD to yield *N*^2^-ADPR-dG in the plasmid, which was then transfected into normal human fibroblast WI38-VA13 for replication. After treatment with pierisin-1, the mutant frequency of the *supF* gene increased 40-fold. The base sequence analysis of these genes identified mutations mainly involving G:C base pairs in both the *HPRT* and *supF* genes, consistent with the fact that pierisin-1 ADP ribosylates guanine bases at an *N*^2^ position in DNA. In the case of the *HPRT* gene, G:C to C:G transversions were the most predominant, followed by G:C to T:A base substitution mutations. In the *supF* gene, G:C to T:A transversions were dominant, followed by G:C to C:G and G:C to A:T transversions. The differences in mutational signatures between the *HPRT* and *supF* genes might be due to cell-specific rather than gene-specific factors such as DNA repair or TLS.

DNA adducts with B[*a*]P and TAM at the *N*^2^ position of the dG residue in DNA are known to predominantly induce G:C to T:A transversions in various DNA targets, including the *HPRT* and *supF* genes, and site-specifically modified oligodeoxynucleotides [37,38,39,40]. Indeed, the mutation signatures of these dG-*N*^2^-adducts are similar to those of pierisin-1, but a considerable number of G:C to C:G base substitutions could be characteristic of pierisin-1 [41], since G:C to C:G transversions are rare mutations [43,44] and only quinoline has been reported to induce G:C to C:G at a high frequency [45]. Mutational hot spots with pierisin-1 were located at G:C base pairs in 5′-rgg-3′ sites in the *supF* gene and in 5′-tgga-3′ or 5′-tggt-3′ in the *HPRT* gene [41].

B[a]P, dG-*N*^2^-BPDE and *N*-(2′-deoxyguanosin-8-yl)-2-aminofluorene adducts can form stable purine–purine base pairings [46,47]. Furthermore, the *N*^2^ modification of guanine drives syn-anti conformational switching and leads to the formation of guanine–guanine mispairing with the *N*^7^- and *O*^6^-positions of the modified guanine for hydrogen bonding [48]. *N*^2^-ADPR-dG might form stable base pairs with guanine or adenine, particularly guanine–guanine mispairing, resulting in G:C to C:G and G:C to T:A mutations [41] (Figure 3).

## 5. Future Directions

*N*^2^-ADPR-dG in double-stranded DNA (ds DNA) formed by pierisin is recognized as DNA damage by the cellular NER system. However, cells have various DNA repair systems in addition to NER. The involvement of other DNA repair systems cannot be excluded at present. A cell viability assay using various DNA repair-deficient cells, for example, may reveal the contribution of other repair systems.

Although the in vitro mutation assays with mammalian cells revealed that *N*^2^-ADPR-dG induces G:C to T:A and C:G base substitutions, the TLS experiments could not identify which polymerase generates G to C and T base substitutions. A two-step model is proposed as a mechanism of the mutagenic bypass of DNA lesions [21,49]. In this model (Figure 2), two DNA polymerases act sequentially, where one polymerase incorporates deoxynucleotides opposite the DNA lesions, and then the other extends the DNA strand from such deoxynucleotides. In TLS across *N*^2^-BPDE-dG, Pol η incorporates dA opposite the lesion, and subsequently, Pol κ extends the DNA chain [50,51,52]. Pol κ has the ability to extend the DNA strand from primer–terminal mispairs opposite the damaged DNA templates [50,53]. In the TLS experiments, Pol η inserted any base opposite *N*^2^-ADPR-dG with a low efficiency, while Pol κ extended the DNA strand from a mismatch terminus opposite *N*^2^-ADPR-dG [32]. Pol κ by itself may not achieve mutagenic TLS across *N*^2^-ADPR-dG, but it may do so together with Pol η, acting as a promiscuous extender of primer–terminal mispairs. Therefore, Pol η and Pol κ might be responsible for the G to T and C mutations. Mutagenesis assays using XPV cells lacking Pol η- and Pol κ-knockout cells should help to understand the involvement of Pol η and Pol κ in the induction of mutation by pierisin-1.

Pierisin has been found as an apoptotic cell death inducer [2,3]. However, the biological significance of mutagenesis by *N*^2^-ADPR-dG in surviving target cells is not clear. The presence of *N*^2^-ADPR-dG in the template strand retards DNA replication. Therefore, pierisin is considered part of the defense system of cabbage butterflies against pathogens by causing replication block in the pathogen genome [54]. On the other hand, there are other DNA-targeting toxins which modify DNA bases. A bacterial cytosine deaminase DddA (double-stranded DNA deaminase A, identified in *Burkholderia cenocepacia*) is one of such toxins [55,56]. Cytosine deamination generates uracil. DddA demonstrates the killing effect, but some target bacterial species resist and accumulate mutations [54,55]. de Moraes et al. indicate that this mutagenic phenomenon could be an evolutionarily selected property. At the moment, there is no definitive study to answer the question: is the mutagenesis ether a main function of pierisin or just an opportunistic outcome? In vivo studies are required to clarify the consequences of mutations induced by*N*^2^-ADPR-dG and the downstream processes of DNA ADP-ribosylation by pierisin.

## 6. Conclusions

While many bacterial mono-ADP-ribosylating toxins modify eukaryotic proteins, the butterfly pierisin mono-ADP-ribosylates guanine residue of DNA, yielding *N*^2^-ADPR-dG. *N*^2^-ADPR-dG in ds DNA is recognized as DNA damage by the cellular NER system. Unrepaired *N*^2^-ADPR-dG can serve as a substrate for TLS, and pol κ and other TLS polymerases could be involved in TLS across *N*^2^-ADPR-dG. Replication past this damaged dG induces G to T and G to C mutations (Figure 4).

## Figures and Tables

**Figure 3 toxins-16-00331-f003:**
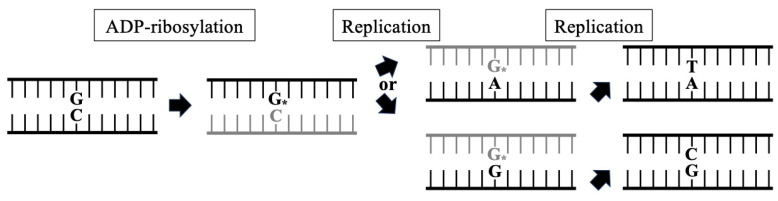
Mutagenesis of ADP-ribosylated dG. Replication past *N*^2^-ADPR-dG (G*) induces G to T or G to C mutations. The template strands of the replication steps are indicated as black lines.

**Figure 4 toxins-16-00331-f004:**
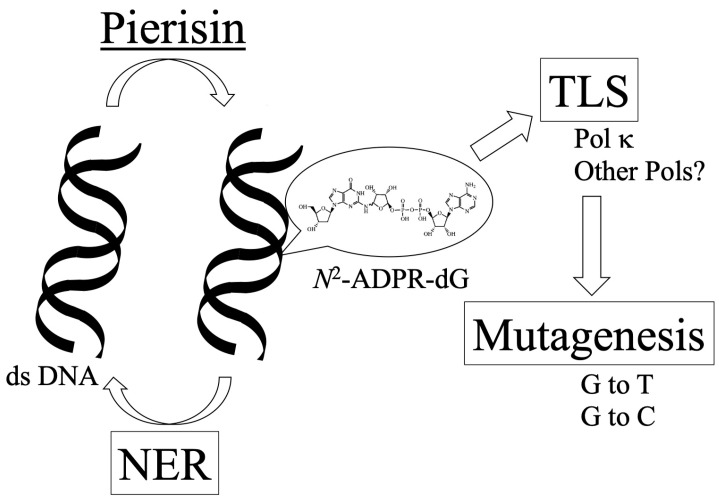
Overview of DNA repair and mutagenesis of *N*^2^-ADPR-dG by pierisin.

## Data Availability

No new data were created.

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
