# Peer review of "DNA Repair and Mutagenesis of ADP-Ribosylated DNA by Pierisin"

_toxins, 2024, doi:10.3390/toxins16080331_

Round 1

Reviewer 1 Report

Comments and Suggestions for Authors

While many bacterial mono-ADP-ribosylating toxins modify eukaryotic proteins, the butterfly pierisins, which are structurally related to bacterial ADP-ribosylating toxins, mono-ADP-ribosylate guanine residues of DNA. This well-written review discusses the functional consequences of guanine modification of DNA by pierisin1. This is of interest and importance, because the physiological or pathophysiological roles of pierisins are still debated.

I have only few comments:

1. It would be helpful, if the scheme of the NER reaction is complemented by proteins involved in NER and mentioned in the text.

2. Similarly, addition of a more detailed scheme of TLS, including some proteins involved would help to follow the text.

3. What is the evidence that no other DNA repair mechanisms like BER are involved in repair of mono-ADP-ribosylated guanine.  

4. It would be nice, if the possible functional mechanism of pierisin-induced ADP-ribosylation would be discussed in some more detail

Author Response

Comment 1: It would be helpful, if the scheme of the NER reaction is complemented by proteins involved in NER and mentioned in the text.

Response 1: Thank you for your suggestion. According to the suggestion, the authors modified Figure 1 and its corresponding texts. We marked in red.

Comment 2: Similarly, addition of a more detailed scheme of TLS, including some proteins involved would help to follow the text.

Response 2: Thank you for your suggestion. According to the suggestion, the authors modified Figure 2 and its corresponding texts. We marked in red.

Comment 3: What is the evidence that no other DNA repair mechanisms like BER are involved in repair of mono-ADP-ribosylated guanine.

Response 3: Involvement of other DNA repair systems cannot be excluded at present. So, the authors added the explanation “N2-ADPR-dG in double-stranded DNA (ds DNA) formed by pierisin is recognized as a DNA damage by the cellular NER system. However, cells have various DNA repair systems in addition to NER. Involvement of other DNA repair systems cannot be excluded at present. Cell viability assay using various DNA repair-deficient cells, for example, may reveal the contribution of other repair systems.” into the beginning of section 5 (Future directions section). We marked them in red.

Comment 4: It would be nice, if the possible functional mechanism of pierisin-induced ADP-ribosylation would be discussed in some more detail.

Response 4: Thank you for your suggestion. In the same special issue of Toxins, Takahashi-Nakaguchi et al. summarize possible functional mechanism of pierisin-induced ADP-ribosylation (Toxins 2024, 16(6), 270). Therefore, the authors refer to the article in the introduction section. Instead, the authors would like to focus on DNA repair, TLS and mutagenesis of N2-ADPR-dG by pierisin-1 in the present manuscript.

Reviewer 2 Report

Comments and Suggestions for Authors

An interesting and comprehensive review, can be published without editing.

The review describes a novel phenomenon, which was recently described: parylated guanine recognized as a DNA lesion. The papaer fill a considerable gap in information about above phenomenon. There are very few publications on thetopic, authors scrupulously describe eksperiments published in the reviewed papers. Conclusions are consistent, references are appropriate. The menuscript is worth publishing without any improvements!

Author Response

Thank you very much for your comments that highly regard our manuscript. According to comments of other reviewers, the manuscript will be modified. I greatly appreciate your review.

Reviewer 3 Report

Comments and Suggestions for Authors

Review comment:Toxins-3064726

DNA repair and mutagenesis of ADP-ribosylated DNA by pierisin

This paper reviewed the effect of Pierisin in DNA repair, TLS in cells and resulting mutagenesis in cell lines. They reviewed how NER repair is involved in Pierisin induced apoptosis. They also reviewed the role of Polymerase k in bypassing N2-ADPR-dG and therefore inducing TLS, and reviewed how Pierisin was shown to induce mutagenesis in mammalian cell lines. In future direction, they suggested further research in discovering which polymerase is involved in mutagenesis inducing DNA replication and clarify if this mutagenesis in a main function in the apoptotic ability of Pierisin toxin.

Overall, while the review was relatively short, it conveys clearly researches done in term of Pierisin effects on DNA damage. As a small correction, we suggest to include nomenclature for the cabbage white butterfly (Pieris brassicae) in abstract.

Major comments:                                                  

1. I can understand that Figure 1. is a process of NER, but I am confused whether this is the ADP-ribosylated DNA NER by Piersin-1 mentioned in paragraph 2 or the general NER process.

2. In line 122, it would be helpful to see a table of the IC50 results for the cell viability assay performed by the authors.

3. In paragraph 4, it would be better to include an additional figure to help understand the explanation.

4. In the future directions section, it is well understood to mention the mechanisms involved in the repair of N2-ADPR-dG by Pierisin and the experimental methods and emphasize the need for further research. However, I feel that there is a lack of prospect and direction on how to connect with existing research, how to conduct further research, and how the results of this research can eventually contribute to solving clinical and medical problems (N2-ADPR-dG-related mutant diseases).

5. This paper is based on experimental results using E. coli and specific mammalian cells (Chinese hamster cells). Are there any results from in vivo studies?

6. While the paper focuses on explaining the mechanisms of NER and TLS, a detailed explanation of interactions with other DNA repair pathways is needed. Additionally, specific examples or case studies of how N2-ADPR-dG is processed under certain conditions are lacking.

7. An explanation of the impact of Pierisin-1 on humans and its association with actual disease occurrence is needed.

8. In the Introduction, it needs to include an overview of current research trends in DNA repair systems. It is also necessary to add explanations about nucleotide excision repair (NER) and translesion DNA synthesis (TLS), which are the main focuses of the paper.

9. It is necessary to further elaborate on the strengths and characteristics of Pierisin-1 in inducing DNA mutations. Adding information about different types and characteristics of ADP-ribosyltransferases, and comparing them with Pierisin-1, would enhance the paper.

10. It needs to include a detailed explanation of the roles and interactions of the various DNA polymerases involved in the TLS process.

11. It needs to describe “Conclusion section”.

12. In the “Future directions” section, it is necessary to describe in detail the limitations and challenges in current pierisin research and propose specific methods to overcome them. Additionally, including an examination of the practical value of pierisin research will strengthen the paper.

13. In the translation DNA synthesis section, the inclusion of experimental results with human DNA polymerases could enhance the section.

Comments on the Quality of English Language

Moderate corrections

Author Response

General Comment: As a small correction, we suggest to include nomenclature for the cabbage white butterfly (Pieris brassicae) in abstract.

Response: Thank you for your suggestion. According to the suggestion, the authors added species name “Pieris rapae” in line 5. We marked it in red.

Comment 1: I can understand that Figure 1 is a process of NER, but I am confused whether this is the ADP-ribosylated DNA NER by Piersin-1 mentioned in paragraph 2 or the general NER process.

Response 1: As the authors have explained in the beginning of the section (DNA repair section) where the Figure 1 is referred, it is the general NER process. NER is general repair process. No special (individual) NER exists for specific DNA damage.

Comment 2: In line 122, it would be helpful to see a table of the IC50 results for the cell viability assay performed by the authors.

Response 2: As the reviewer’s indication, the authors added Table 1 of IC50 and referred as “They showed that IC50 values of pierisin-1 were 650 ng/ml for AA8 (wild) and 190-260 ng/ml for NER-deficient cells (Table 1).” in the main text (line 127-8). We marked in red.

Comment 3: In paragraph 4, it would be better to include an additional figure to help understand the explanation.

Response 3: Thank you for your suggestion. According to the suggestion, the authors added Figure 3 and referred it in the end of the section 4 (mutagenesis section). We marked its legend in red.

Comment 4: In the future directions section, it is well understood to mention the mechanisms involved in the repair of N2-ADPR-dG by Pierisin and the experimental methods and emphasize the need for further research. However, I feel that there is a lack of prospect and direction on how to connect with existing research, how to conduct further research, and how the results of this research can eventually contribute to solving clinical and medical problems (N2-ADPR-dG-related mutant diseases).

Response 4: Pierisin is NOT a pathogen against human beings.

Comment 5: This paper is based on experimental results using E. coli and specific mammalian cells (Chinese hamster cells). Are there any results from in vivo studies?

Response 5: No in vivo study on mutagenesis by pierisin has been reported.

Comment 6: While the paper focuses on explaining the mechanisms of NER and TLS, a detailed explanation of interactions with other DNA repair pathways is needed. Additionally, specific examples or case studies of how N2-ADPR-dG is processed under certain conditions are lacking.

Response 6: There is a quite good and comprehensive review on DNA damage repair and mutagenesis by Chatterjee, N., et al., therefore the authors refer the article as ref# 7 in the introduction section. Involvement of other DNA repair systems cannot be excluded at present. So, the authors added the explanation “N2-ADPR-dG in double-stranded DNA (ds DNA) formed by pierisin is recognized as a DNA damage by the cellular NER system. However, cells have various DNA repair systems in addition to NER. Involvement of other DNA repair systems cannot be excluded at present. Cell viability assay using various DNA repair-deficient cells, for example, may reveal the contribution of other repair systems.” into the “5. Future directions” section. We marked them in red.

Comment 7: An explanation of the impact of Pierisin-1 on humans and its association with actual disease occurrence is needed.

Response 7: Pierisin is not a human pathogen.

Comment 8: In the Introduction, it needs to include an overview of current research trends in DNA repair systems. It is also necessary to add explanations about nucleotide excision repair (NER) and translesion DNA synthesis (TLS), which are the main focuses of the paper.

Response 8: DNA repair and mutagenesis forms quite broad range of research field. Therefore, the authors think it is better to cite the comprehensive review article by Chatterjee, N., et al., (as ref# 7) than to state the overview for ourselves in the present manuscript. In fact, the reviewers 1 and 2 evaluate this concise story of the present manuscript highly. Furthermore, in each section (sections 2,3,4) the authors provide general but more detailed information of each step. If necessary, appropriate review articles are referred, for example, excellent reviews (more specialized on TLS) by Goodman, M.F., et al. are also referred as refs# 17, 49 in the manuscript.

Comment 9: It is necessary to further elaborate on the strengths and characteristics of Pierisin-1 in inducing DNA mutations. Adding information about different types and characteristics of ADP-ribosyltransferases, and comparing them with Pierisin-1, would enhance the paper.

Response 9: There is no study on direct outcome of mutagenesis by pierisin. As mentioned by Takahashi-Nakaguchi et al.in Toxins 2024, 16(6), 270 (our ref#6), primary role of ADP-ribosylation by pierisin could be to induce replication arrest and apoptosis in the target cells. Furthermore, target of many other ADP-ribosyltransferases is protein, not DNA. Instead, the authors refer other DNA-targeting toxins modifying DNA bases (cytosine deaminase DddA) in the end of the section 5 (Future directions section).

Comment 10: It needs to include a detailed explanation of the roles and interactions of the various DNA polymerases involved in the TLS process.

Response 10: No such study on TLS by pierisin-induced DNA lesion has been reported. Therefore, studies on TLS over DNA lesion with modification at 2-amino-group of dG, same as pierisin-induced DNA damage, have been stated in the 2nd paragraph of the section 5 (Future directions section).

Comment 11: It needs to describe “Conclusion section”.

Response 11: According to the reviewer’s direction, the authors added the Conclusion section in the end of the manuscript and marked in red.

Comment 12: In the “Future directions” section, it is necessary to describe in detail the limitations and challenges in current pierisin research and propose specific methods to overcome them. Additionally, including an examination of the practical value of pierisin research will strengthen the paper.

Response 12: The authors added the description of limitations and challenges of DNA repair topic into the beginning of section 5 (Future directions section). We marked them in red. Limitations and challenges on the mutagenesis have been mentioned in the ends of 2nd and 3rdparagraphs of the section 5.

Comment 13: In the translation DNA synthesis section, the inclusion of experimental results with human DNA polymerases could enhance the section.

Response 13: In the 2nd paragraph of the section 3 (TLS section), the authors have explained experimental results of human DNA polymerases.

Round 2

Reviewer 3 Report

Comments and Suggestions for Authors

accept